# QT-ViT: Improving Linear Attention in ViT with Quadratic Taylor Expansion

**Yixing Xu, Chao Li, Dong Li, Xiao Sheng, Fan Jiang, Lu Tian, Emad Barsoum**
Advanced Micro Devices, Inc., Beijing, China
{yixing.xu, chao.li, d.li, xsheng, f.jiang, lu.tian, emad.barsoum}@amd.com

## Abstract

Vision transformer model (ViT) is widely used and performs well in vision tasks due to its ability to capture long-range dependencies. However, the time complexity and memory consumption increase quadratically with the number of input patches which limits the usage of ViT in real-world applications. Previous methods have employed linear attention to mitigate the complexity of the original self-attention mechanism at the expense of effectiveness. In this paper, we propose QT-ViT models that improve the previous linear self-attention using quadratic Taylor expansion. Specifically, we substitute the softmax-based attention with second-order Taylor expansion, and then accelerate the quadratic expansion by reducing the time complexity with a fast approximation algorithm. The proposed method capitalizes on the property of quadratic expansion to achieve superior performance while employing linear approximation for fast inference. Compared to previous studies of linear attention, our approach does not necessitate knowledge distillation or high-order attention residuals to facilitate the training process. Extensive experiments demonstrate the efficiency and effectiveness of the proposed QT-ViTs, showcasing the state-of-the-art results. Particularly, the proposed QT-ViTs consistently surpass the previous SOTA EfficientViTs under different model sizes, and achieve a new Pareto-front in terms of accuracy and speed.

## 1 Introduction

Compared to convolutional neural networks (CNNs), vision transformers (ViTs) are getting more and more attention due to their strong performance across various computer vision tasks, such as image classification [33, 16, 8, 37, 38, 21], object detection [4, 13], semantic segmentation [36, 15] and low-level vision [20, 32, 31, 30]. The effectiveness of ViT comes from the multi-head self-attention (MHSA) operation that allows the model to capture long-range information by calculating the attention score between each pair of patches. However, this mechanism necessitates quadratic time and storage complexity $\mathcal{O}(n^2)$ related to the number of input patches $n$, and the original ViTs require significant computational and storage resources when applied to real-world applications.

To overcome the aforementioned problem, previous researches focus on improving the original self-attention mechanism by using local attention such as window attention [23], dilated attention [10] and random attention [27]. Another family of methods is to utilize linear attention [11, 2, 5, 3] that decomposes the original softmax function into two non-linear kernels so that the order of matrix multiplications in attention score calculation is changed to reduce the quadratic computational complexity into a linear one. Many papers focus on designing non-linear kernels and novel linear-attention architectures for better approximation, *e.g.,* Hydra-attention [2] uses the hydra trick to their multi-head attention by setting as many heads as features. Performer [5] uses fast attention via a positive orthogonal random features approach to approximate the softmax attention. EfficientViT [3] replaces softmax with ReLU non-linear activation and applies depthwise and group convolution to

38th Conference on Neural Information Processing Systems (NeurIPS 2024).

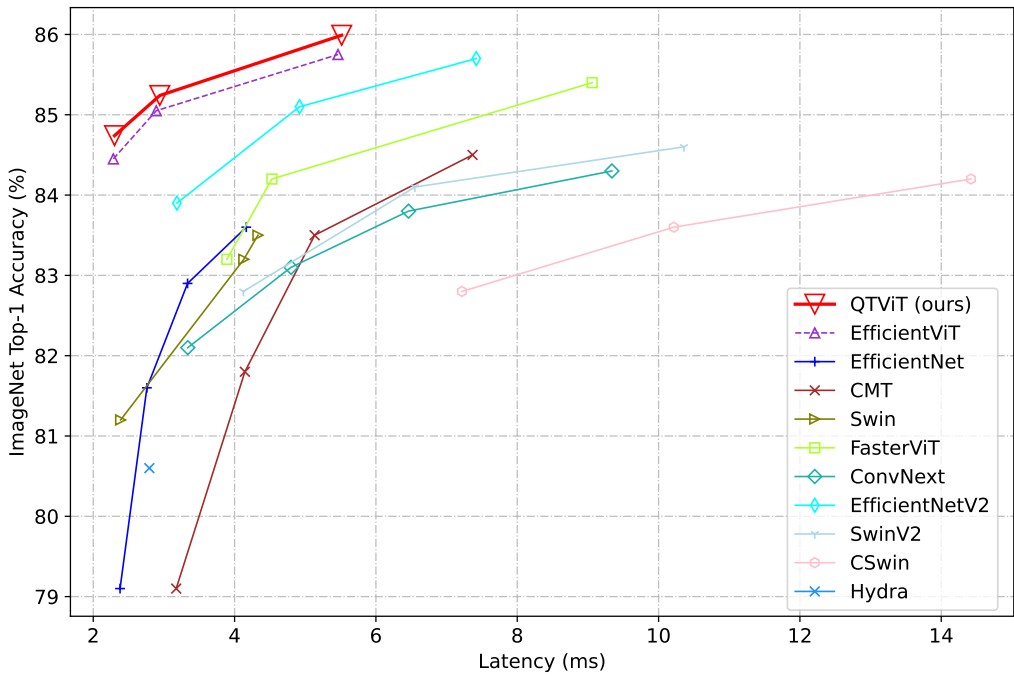

Figure 1: The accuracy-speed trade-offs of the proposed QT-ViTs and other state-of-the-art transformer models on the ImageNet dataset. Latencies are evaluated on the AMD Instinct MI250 GPU.

improve its performance. Flatten transformer [14] utilizes focused attention based on ReLU to force the attention operation to focus on more informative regions.

Previous linear attention methods reduce the complexity of the attention mechanism from $\mathcal{O}(n^2 d)$ to $\mathcal{O}(nd^2)$ at the expense of the performance on visual tasks, where $d$ is the patch dimensionality. Some of them necessitate the knowledge distillation method [7] or high-order attention residuals [39] to make up for the performance gap. However, the GPU memory consumption will severely increase which makes these methods unsuitable for training large transformer models.

In this paper, we explore the utilization of second-order (quadratic) Taylor expansion to approximate the original softmax attention. We theoretically show that this approximation can be decomposed into two non-linear kernels through the utilization of the Kronecker product [34]. By employing this approach, the computational complexity can be changed from $\mathcal{O}(n^2 d)$ to $\mathcal{O}(nd^3)$. We then propose a fast approximation algorithm to accelerate the computation of the Kronecker product, thereby reducing the complexity to $\mathcal{O}(nd^2)$. In contrast to the first-order (linear) Taylor expansion [7] and other linear attention methods, we can utilize the high-order information within the softmax function to achieve superior performance while at the same time preserving the efficiency of linear attention. Experimental results on the ImageNet dataset show that the proposed QT-ViTs can achieve a superior accuracy-speed trade-off when compared to other state-of-the-art methods, as shown in Fig. 1. Additionally, we conduct experiments on object detection and semantic segmentation tasks to further validate the effectiveness of our approach.

## 2 Preliminaries

In this section, we first introduce the preliminaries of softmax attention and linear attention. Then, we provide an overview of various instantiations of the original linear attention method used in ViT and analyze their advantages and disadvantages.

## 2.1 Softmax Self-Attention

Softmax self-attention operation is the key component in the transformer model. Given an input matrix $\mathbf{X} \in \mathbb{R}^{N \times d}$ where $N$ is the number of patches and $d$ is the dimension of each patch, we first map the input matrix to the query, key and value embeddings by using the matrix multiplications:

$$\mathbf{Q} = \mathbf{X}\mathbf{W}_Q, \quad \mathbf{K} = \mathbf{X}\mathbf{W}_K, \quad \mathbf{V} = \mathbf{X}\mathbf{W}_V, \tag{1}$$

where $\mathbf{W}_Q$, $\mathbf{W}_K$ and $\mathbf{W}_V \in \mathbb{R}^{d \times d}$ are learnable matrices. Then, the attention score is computed on each pair of patches to capture the global information as shown below:

$$\mathbf{O}_k = \sum_{i=1}^{N} \frac{\mathrm{Sim}(\mathbf{Q}_k, \mathbf{K}_i)}{\sum_{j=1}^{N} \mathrm{Sim}(\mathbf{Q}_k, \mathbf{K}_j)} \mathbf{V}_i = \sum_{i=1}^{N} \frac{\exp(\mathbf{Q}_k \mathbf{K}_i^{\top}/\sqrt{d})}{\sum_{j=1}^{N} \exp(\mathbf{Q}_k \mathbf{K}_j^{\top}/\sqrt{d})} \mathbf{V}_i, \tag{2}$$

where $\mathrm{Sim}(\mathbf{Q}_k, \mathbf{K}_i) = \exp(\mathbf{Q}_k \mathbf{K}_i^{\top}/\sqrt{d})$ is the similarity measurement function in the softmax attention, $\mathbf{Q}_k$, $\mathbf{K}_i$ ($\mathbf{K}_j$), $\mathbf{V}_i$, $\mathbf{O}_k$ are the corresponding $k$-th, $i$-th ($j$-th), $i$-th, $k$-th row vectors of the query, key, value and output matrices, respectively. The inner product of the query-key pair is first computed to calculate the similarity between the pair, then a scale is applied for stability and a softmax function is used to transfer the similarity into probability. This probability is applied to the value matrix to get the final attention score output. The softmax attention computes the inner products of all the query-key pairs and results in a $\mathcal{O}(N^2 d)$ time complexity.

## 2.2 Linear Self-Attention

The overhead of the computation of Eq. 2 mainly comes from the matrix multiplication. By decomposing the similarity function into two separate kernel embeddings, *i.e.,* $\mathrm{Sim}(\mathbf{Q}_k, \mathbf{K}_i) = \phi(\mathbf{Q}_k)\phi(\mathbf{K}_i)^{\top}$, and the original softmax attention function can be changed into linear attention by exchanging the order of matrix multiplication:

$$\mathbf{O}_k = \sum_{i=1}^{N} \frac{\phi(\mathbf{Q}_k)\phi(\mathbf{K}_i)^{\top}}{\sum_{j=1}^{N} \phi(\mathbf{Q}_k)\phi(\mathbf{K}_j)^{\top}} \mathbf{V}_i = \frac{\phi(\mathbf{Q}_k)\left(\sum_{i=1}^{N} \phi(\mathbf{K}_i)^{\top} \mathbf{V}_i\right)}{\phi(\mathbf{Q}_k)\left(\sum_{j=1}^{N} \phi(\mathbf{K}_j)^{\top}\right)}, \tag{3}$$

where the complexity is changed from $\mathcal{O}(N^2 d)$ to $\mathcal{O}(N d^2)$. Since the patch dimension $d$ is always smaller than the number of patches $N$ in the popular ViT architectures, the computation overhead can thus be reduced.

However, in order to losslessly decompose the similarity function in the softmax attention $\mathrm{Sim}(\mathbf{Q}_k, \mathbf{K}_i)$ into the product of two kernel embeddings $\phi(\mathbf{Q}_k)$ and $\phi(\mathbf{K}_i)$, the dimensionality of the kernel function needs to be infinite which is unable to apply to real-world applications. Thus, a series of instantiations are proposed trying to compute $\phi(\cdot)$ efficiently while preserving as much information of the original similarity function as possible.

In the following, we use $\mathbf{q} \triangleq \mathbf{Q_k}$ and $\mathbf{k} \triangleq \mathbf{K_i}$ to represent row vectors in query matrix $\mathbf{Q}$ and key matrix $\mathbf{K}$ that do not belong to any specific row.

## 2.3 Instantiations of the Kernel Function

Linear transformer was first proposed in [19] and $\phi(\mathbf{x}) = \mathrm{elu}(\mathbf{x}) + 1$ was used as the kernel function. EfficientViT [3] used $\phi(\mathbf{x}) = \mathrm{ReLU}(\mathbf{x})$ as the instantiation. Both methods ensure that $\phi(\mathbf{q})\phi(\mathbf{k})^{\top} \geq 0$ which is consistent with the property of the similarity measurement function $\mathrm{Sim}(\cdot)$. Flatten Transformer [14] argued that previous approximations smooth the distribution of linear attention which failed to focus on more informative regions, and proposed a focused function $\phi_p(\mathbf{x}) = \frac{||\mathrm{ReLU}(\mathbf{x})||}{||\mathrm{ReLU}(\mathbf{x})^{**p}||} \mathrm{ReLU}(\mathbf{x})^{**p}$ where $||\cdot||$ represents the Euclidean norm and $(\cdot)^{**p}$ is element-wise power $p$ of the input. Hydra attention [2] used cosine similarity as the kernel $\phi(\mathbf{x}) = \mathbf{x}/||\mathbf{x}||_2$, PolyNL [1] used mean kernel $\phi(\mathbf{x}) = \mathbf{x}/\sqrt{N}$, and AFT-Simple [41] proposed different functions for query $\phi(\mathbf{q}) = \mathrm{sigmoid}(\mathbf{q})$ and key $\phi(\mathbf{k}) = \mathrm{softmax}(\mathbf{k})$, respectively. These methods suffered from the performance drop since they lacked sufficient expression ability to replicate the original softmax attention mechanism.

Besides the aforementioned methods, some studies approximated the similarity function with kernel expansions such as angular kernel expansion [39] with $\mathrm{Sim}(\mathbf{q}, \mathbf{k}) = 1/2 + 1/\pi \cdot (\mathbf{q}\mathbf{k}^\top) + H_r$ or first order Taylor expansion [7] with $\mathrm{Sim}(\mathbf{q}, \mathbf{k}) = 1 + \mathbf{q}\mathbf{k}^\top/\sqrt{d} + H_r$ where $H_r$ represents the high-order residuals. These methods necessitated the masked output of original softmax attention as $H_r$ and applied the knowledge distillation (KD) method to further enhance the performance which severely increased the GPU memory consumption and were unsuitable for training large transformer models.

# 3   Methods

In this section, we propose to use second-order (quadratic) Taylor expansion to approximate the similarity measurement function $\mathrm{Sim}(\cdot, \cdot)$ in Eq. 2. Compared to the first-order (linear) Taylor expansion [7], quadratic approximation contains less information in the high-order residuals. Therefore, we can directly ignore them and derive a good performance without utilizing masked softmax attention output or the KD method.

However, it is non-trivial to decompose the quadratic Taylor expansion into separate kernel embeddings with linear time complexity. Thus, in the following we first give a theoretical derivation by using the Kronecker product to decompose the quadratic expansion. Then, a fast approximation algorithm is applied to accelerate the computation of the Kronecker product.

## 3.1   Decompose Quadratic Taylor Expansion

The quadratic Taylor expansion of the similarity measurement function is expressed as:

$$
\begin{aligned}
\mathrm{Sim}(\mathbf{q}, \mathbf{k}) = \exp\left(\frac{<\mathbf{q}, \mathbf{k}>}{\sqrt{d}}\right) &\approx 1 + \frac{<\mathbf{q}, \mathbf{k}>}{\sqrt{d}} + \frac{<\mathbf{q}, \mathbf{k}>^2}{2d} \\
&= \frac{\left(\frac{<\mathbf{q}, \mathbf{k}>}{\sqrt{d}} + 1\right)^2 + 1}{2} \\
&= \frac{<\phi(\mathbf{q}), \phi(\mathbf{k})>^2 + 1}{2},
\end{aligned}
\tag{4}
$$

where $< \cdot, \cdot >$ is the dot product and $\phi(\mathbf{x}) = \left[\frac{\mathbf{x}}{\sqrt[4]{d}}, 1\right]$ is used for vectors $\mathbf{q}$ and $\mathbf{k}$. However, since the quadratic term exists in Eq. 4, it is challenging to decompose the equation into two separate kernel embeddings. In the following, we show that this problem can be solved by using the Kronecker product.

Given two vectors $\mathbf{a} = \{a_i\}_{i=1}^d$ and $\mathbf{b} = \{b_i\}_{i=1}^d$, we can easily derive:

$$
<\mathbf{a}, \mathbf{b}>^2 = \left(\sum_{i=1}^d a_i b_i\right)^2 = \sum_{i=1}^d a_i^2 b_i^2 + 2 \sum_{i=1}^{d-1} \sum_{j=i+1}^d a_i b_i a_j b_j.
\tag{5}
$$

This is equal to first computing the Kronecker product of each vector and then applying dot product, *i.e.*, given $K_r(\mathbf{x}) = \mathrm{vec}(\mathbf{x} \otimes \mathbf{x})$ where $\otimes$ represents the Kronecker product and $\mathrm{vec}(\cdot)$ is the vectorized output, we have:

$$
\begin{aligned}
<K_r(\mathbf{a}), K_r(\mathbf{b})> &= [a_1 \mathbf{a}, \cdots, a_d \mathbf{a}] \cdot [b_1 \mathbf{b}, \cdots, b_d \mathbf{b}] \\
&= [a_1 a_1, \cdots, a_1 a_d, a_2 a_1, \cdots, a_2 a_d, \cdots, a_d a_1, \cdots, a_d a_d] \\
&\quad \cdot [b_1 b_1, \cdots, b_1 b_d, b_2 b_1, \cdots, b_2 b_d, \cdots, b_d b_1, \cdots, b_d b_d] \\
&= \sum_{i=1}^d a_i^2 b_i^2 + 2 \sum_{i=1}^{d-1} \sum_{j=i+1}^d a_i b_i a_j b_j \\
&= <\mathbf{a}, \mathbf{b}>^2.
\end{aligned}
\tag{6}
$$

Then, we can apply Eq. 6 to Eq. 4 and decompose the similarity function into two separate kernel embeddings:

$$
\begin{aligned}
\mathrm{Sim}(\mathbf{q}, \mathbf{k}) &\approx \frac{<\phi(\mathbf{q}), \phi(\mathbf{k})>^2 + 1}{2} \\
&= \frac{<K_r(\phi(\mathbf{q}), K_r(\phi(\mathbf{k}))> + 1}{2} \\
&= <\varphi(\mathbf{q}), \varphi(\mathbf{k})>,
\end{aligned}
\tag{7}
$$

where

$$
\varphi(\mathbf{x}) = \left[ \frac{1}{\sqrt{2}} K_r(\phi(\mathbf{x})), \frac{1}{\sqrt{2}} \right] = \left[ \frac{1}{\sqrt{2}} \mathrm{vec}(\phi(\mathbf{x}) \otimes \phi(\mathbf{x})), \frac{1}{\sqrt{2}} \right]
\tag{8}
$$

is the kernel function applied to the query and key vectors. Note that given a vector $\mathbf{x} \in \mathbb{R}^d$, Kronecker product gives an output vector with quadratic length $K_r(\mathbf{x}) \in \mathbb{R}^{d^2}$. Thus, the time complexity of linear attention using the decomposed quadratic Taylor expansion is $\mathcal{O}(Nd^3)$. Compared to the original softmax attention with $\mathcal{O}(N^2 d)$ time complexity, the proposed method does not yield an advantage.

## 3.2 Reduce the Time Complexity

Recall that the computational burden primarily arises from the Kronecker product that quadratically expands the input dimension. Thus, there are several simple ways to reduce the dimension. For example, a pooling function can be applied on the input vector $\mathbf{y} = \mathrm{pool}(\mathbf{x}) \in \mathbb{R}^{d/p}$ where $p$ is the dimensionality reduction factor. The output dimension of the Kronecker product can be reduced to $K_r(\mathbf{y}) \in \mathbb{R}^{d^2/p^2}$, and the corresponding time complexity of the linear attention is $\mathcal{O}(Nd^3/p^2)$. Another way is to divide the input vector into $c$ chunks $\mathbf{x} = [\mathbf{x}^1, \cdots, \mathbf{x}^c]$ and compute the Kronecker product within each chunk $K_r(\mathbf{x}^i) \in \mathbb{R}^{d^2/c^2}$, and finally concatenate them together to derive the output $\mathbf{o} = \mathrm{concat}\left( K_r(\mathbf{x}^1), \cdots, K_r(\mathbf{x}^c) \right) \in \mathbb{R}^{d^2/c}$. The time complexity of the linear attention using this method is $\mathcal{O}(Nd^3/c)$.

Although methods mentioned above can decrease the computational load, they do not actually reduce the time complexity. In the following, we propose a fast approximation algorithm to accelerate the computation of the Kronecker product, and reduce the computational complexity from $\mathcal{O}(Nd^3)$ to $\mathcal{O}(Nd^2)$.

By rewriting the definition of $K_r(\phi(\mathbf{x}))$ in Eq. 8 in its element-wise form, we can get:

$$
\begin{aligned}
K_r(\phi(\mathbf{x})) &= K_r([\frac{\mathbf{x}}{\sqrt[4]{d}}, 1]) \\
&= \left[ \frac{x_1}{\sqrt[4]{d}} \cdot [\frac{\mathbf{x}}{\sqrt[4]{d}}, 1], \ \cdots, \ \frac{x_d}{\sqrt[4]{d}} \cdot [\frac{\mathbf{x}}{\sqrt[4]{d}}, 1], \ [\frac{\mathbf{x}}{\sqrt[4]{d}}, 1] \right] \\
&= \left[ \{\frac{x_1 x_1}{\sqrt{d}}, \cdots, \frac{x_1 x_d}{\sqrt{d}}, \frac{x_1}{\sqrt[4]{d}}\}, \cdots, \{\frac{x_d x_1}{\sqrt{d}}, \cdots, \frac{x_d x_d}{\sqrt{d}}, \frac{x_d}{\sqrt[4]{d}}\}, \{\frac{x_1}{\sqrt[4]{d}}, \cdots, \frac{x_d}{\sqrt[4]{d}}, 1\} \right].
\end{aligned}
\tag{9}
$$

Note that the order of the elements in the above equation does not influence the result of the inner product $< K_r(\phi(\mathbf{q}), K_r(\phi(\mathbf{k})) >$ in Eq. 7 as long as $K_r(\phi(\mathbf{q}))$ and $K_r(\phi(\mathbf{k}))$ change the order of their elements in the same manner. Thus, Eq. 9 can be written as:

$$
\widehat{K}_r(\phi(\mathbf{x})) = \mathrm{concat}\left( \frac{\{x_i x_j\}_{i,j=1}^d}{\sqrt{d}}, \frac{\{x_i\}_{i=1}^d}{\sqrt[4]{d}}, \frac{\{x_i\}_{i=1}^d}{\sqrt[4]{d}}, 1 \right),
\tag{10}
$$

which is divided into four terms. The first is the quadratic term that contains $d^2$ components representing the multiplication of each pair of elements in $\mathbf{x}$ (including self-multiplication), the second and third terms are the linear term with length $d$ each, and the fourth term is the constant term. Since the computational load of the inner product in Eq. 10 mainly comes from the quadratic term, it is important to reduce the number of elements in this term. Randomly preserving $d$ items from $d^2$ elements is an efficient approach but leads to poor results. Employing grouping techniques help selecting the most representative items at the cost of increasing the computational complexity

compared to random selection. We empirically find that using the self-multiplication terms $\{x_i^2\}_{i=1}^d$ can effectively represent all quadratic terms, while at the same time maintaining high efficiency.

Therefore, the Kronecker product in Eq. 10 can be replaced with a compact version:

$$\widetilde{K}_r(\phi(\mathbf{x})) = \text{concat}\left(\alpha \cdot \sqrt{d}\frac{\{x_i^2\}_{i=1}^d}{\sqrt{d}}, \beta \cdot \sqrt{2}\frac{\{x_i\}_{i=1}^d}{\sqrt[4]{d}}, \gamma\right)$$

$$= \text{concat}\left(\alpha \cdot \{x_i^2\}_{i=1}^d, \beta \cdot \sqrt[4]{\frac{4}{d}}\{x_i\}_{i=1}^d, \gamma\right), \qquad (11)$$

in which we merge items of the same kind and multiply them by the square root of the number of the same items so as not to affect the inner-product result in Eq. 7. Learnable scalar parameters $\alpha$, $\beta$ and $\gamma$ are used as the trade-off parameters. Note that given a vector $\mathbf{x} \in \mathbb{R}^d$, this compact version of the Kronecker product gives an output of length $2d + 1$. Therefore, the time complexity of linear attention using the decomposed quadratic Taylor expansion is reduced from $\mathcal{O}(Nd^3)$ to $\mathcal{O}(Nd^2)$. We further found that the linear term can be discarded without hurting the classification performance, thus we set $\beta = 0$ in the following experiments.

## 4 Experiments

In this section, we apply our linear attention with quadratic Taylor expansion to vision transformers and propose a series of QT-ViT models. We empirically investigate the effectiveness and efficiency of the proposed models on the ImageNet-1k classification dataset. Additional results regarding the performance on object detection and semantic segmentation tasks are provided in the appendices.

### 4.1 Image Classification

**Datasets and model architectures.** The ImageNet-1k classification dataset is used for training and evaluation, which contains 1.28M training images and 50K validation images from 1000 different classes. We utilize the model architecture proposed in EfficientViT [3] and replace the kernel function with our proposed compact quadratic Taylor expansion kernel. An absolute positional embedding is added to the key matrix before applying linear attention, and a non-linear shortcut $o = o + \text{GELU}(\text{BN}(\text{v}))$ is added to the output of the linear attention $o$ where $v$ is the value matrix. Different exponential moving average (EMA) decay parameters are used, and all the other training settings and hyper-parameters remain the same.

**Compared methods and evaluation metrics.** To verify the effectiveness of the proposed QT-ViTs, we compare our method with a series of competitors including (1) Vision transformers with linear attention such as ViTALiTy [7], Castling-ViT [39], EfficientViT [3], FLatten Transformer [14] and Hydra Attention ViT [2]; (2) Other vision transformers with sparse attention or hierarchical architectures such as Swin [23], SwinV2 [22], FasterViT [17], PoolFormer [40], MobileViT [25], MobileViTV2 [26] and CSwin [9]; (3) State-of-the-art CNN models and CNN-Transformer combined model architectures such as CoAtNet [6], CMT [12], ConvNeXt [24], EfficientNet [28] and EfficientNetV2 [29].

The proposed QT-ViTs and other baseline models are evaluated based on the accuracy-speed trade-offs as shown in Fig. 1. Furthermore, we measure the classification performance with top-1/top-5 accuracy. The efficiency of the model is represented by the FLOPs and parameters. Finally, we evaluate the inference speed of the models on the AMD Instinct MI250 GPU in Fig. 1.

**Experimental results.** The effectiveness and efficiency of the proposed QT-ViTs are evaluated on the ImageNet-1k dataset by comparing them to other state-of-the-art baseline methods mentioned above. The results are shown in Tab. 1 and all methods are gathered by their FLOPs into five groups including: <1G, 1~3G, 3~5G, 5~10G and >10G.

As shown in the table, the proposed QT-ViTs achieve new SOTA accuracy-efficiency trade-off across different FLOPs range. For example, we outperform ViTALiTy who uses the first-order Taylor expansion by a large margin without using knowledge distillation or high-order residuals that severely increase the GPU memory consumption during training. Compared to vision transformer with sparse attention such as CSWin, our QT-ViT-4 achieves 84.7% top-1 accuracy with only 5.26G FLOPs while CSwin-B has 84.2% top-1 accuracy with 15.00G FLOPs, which means that we have 0.5% higher top-1

Table 1: Image classification results on ImageNet-1k dataset. QT-ViTs are compared with SOTA baselines. Methods are grouped based on FLOPs.

| FLOPs range | Model Architecture | Parameters (M) | FLOPs (G) | Top-1 Acc (%) | Top-5 Acc (%) |
|---|---|---|---|---|---|
| <1G | ViTALiTy-DeiT-T [7] | - | 0.33 | 71.9 | - |
| | EfficientNet-B1 [28] | 7.8 | 0.70 | 79.1 | 94.4 |
| | PoolFormer-S12 [40] | 11.9 | 1.82 | 77.2 | - |
| | EfficientViT-B1 [3] | 9.1 | 0.52 | 79.4 | 94.3 |
| | CMT [12] | 9.5 | 0.60 | 79.1 | 94.5 |
| | MobileViT-XS [25] | 2.3 | 0.70 | 74.8 | 92.3 |
| | MobileViTV2-0.5 [26] | 1.4 | 0.50 | 70.2 | - |
| | QT-ViT-1 (ours) | 9.4 | 0.52 | **79.6** | 94.7 |
| 1∼3G | Castling-DeiT-T [39] | 5.6 | 1.18 | 76.0 | 92.5 |
| | EfficientNet-B3 [28] | 12.0 | 1.80 | 81.6 | 95.7 |
| | EfficientViT-B2 [3] | 24.3 | 1.60 | 82.1 | 95.8 |
| | FLatten-PVT-T [14] | 12.2 | 2.00 | 77.8 | - |
| | QT-ViT-2 (ours) | 24.9 | 1.60 | **82.5** | 95.9 |
| 3∼5G | PoolFormer-S24 [40] | 21.4 | 3.40 | 80.3 | - |
| | EfficientNet-B4 [28] | 19.0 | 4.20 | 82.9 | 96.4 |
| | Swin-T [23] | 29.0 | 4.50 | 81.3 | 95.5 |
| | EfficientViT-B3 [3] | 49.0 | 4.00 | 83.5 | 96.4 |
| | FasterViT-1 [17] | 53.4 | 5.30 | 83.2 | 96.5 |
| | ConvNeXt-T [24] | 29.0 | 4.50 | 82.1 | - |
| | QT-ViT-3 (ours) | 49.7 | 3.97 | **83.9** | 96.7 |
| 5∼10G | PoolFormer-M36 [40] | 56.2 | 8.78 | 82.1 | - |
| | EfficientNet-B5 [28] | 30.0 | 9.90 | 83.6 | 96.7 |
| | EfficientNetV2-S [29] | 22.0 | 8.40 | 83.9 | - |
| | SwinV2-T [22] | 28.0 | 6.60 | 82.8 | - |
| | EfficientViT-L1 [3] | 53.0 | 5.30 | 84.5 | 96.9 |
| | EfficientViT-L2 [3] | 64.0 | 6.96 | 85.1 | 97.0 |
| | Castling-MViTv2-B [39] | 51.9 | 9.82 | 85.0 | 97.2 |
| | FasterViT-2 [17] | 75.9 | 8.70 | 84.2 | 96.8 |
| | QT-ViT-4 (ours) | 53.0 | 5.26 | 84.7 | 96.7 |
| | QT-ViT-5 (ours) | 64.1 | 6.96 | **85.2** | 97.0 |
| >10G | PoolFormer-M48 [40] | 73.5 | 11.56 | 82.5 | - |
| | CSWin-B [9] | 78.0 | 15.00 | 84.2 | - |
| | EfficientViT-L3 [3] | 246.0 | 28.00 | 85.8 | 97.2 |
| | Castling-DeiT-B [39] | 87.2 | 17.28 | 84.2 | - |
| | Hydra-DeiT-B [2] | - | 17.46 | 80.6 | - |
| | FLatten-CSwin-B [14] | 75.0 | 15.00 | 84.5 | - |
| | SwinV2-B [22] | 88.0 | 21.80 | 84.6 | - |
| | CoAtNet-3 [6] | 168.0 | 35.00 | 84.5 | - |
| | FasterViT-4 [17] | 424.6 | 36.60 | 85.4 | 97.3 |
| | QT-ViT-6 (ours) | 246.8 | 27.60 | **86.0** | 97.3 |

accuracy with 64.9% less FLOPs. For CNN competitors, the QT-ViT-3 outperforms ConvNeXt-T by 1.8% top-1 accuracy with 11.8% less FLOPs. Finally, compared to the current state-of-the-art vision transformer model, the proposed QT-ViT-1∼6 outperforms EfficientViT-B1∼B3 & L1∼L3 by 0.2%, 0.4%, 0.4%, 0.2%, 0.1%, 0.2% respectively with roughly the same FLOPs and parameters. The accuracy-speed trade-offs of the proposed QT-ViTs and other models are shown in Fig. 1.

Table 2: Results of using different kernels. The baseline method uses the original self-attention operation with $\mathcal{O}(N^2 d)$ computational complexity and is used as the strong baseline. Other methods use different linear attentions.

| Method | Kernel | $\phi(\mathbf{q})$ | $\phi(\mathbf{k})$ | Top-1 Acc (%) |
|---|---|---|---|---|
| baseline | - | - | - | 79.8 |
| EfficientViT [3] | ReLU non-linearity | \multicolumn{2}{c}{$\text{ReLU}(x)$} | 79.4 |
| Hydra [2] | cosine similarity | \multicolumn{2}{c}{$x/\|x\|_2$} | 79.1 |
| PolyNL [1] | mean | \multicolumn{2}{c}{$x/\sqrt{N}$} | 78.8 |
| AFT-Simple [41] | sigmoid & softmax | $\sigma(x)$ | $\text{softmax}(x)$ | 78.9 |
| Castling-ViT [39] | angular kernel | \multicolumn{2}{c}{$\text{Sim}(\mathbf{q},\mathbf{k}) = \frac{1}{2} + \frac{1}{\pi} \cdot (\mathbf{q}\mathbf{k}^\top)$} | 79.1 |
| ViTALiTy [7] | 1st order Taylor expansion | \multicolumn{2}{c}{$[x/\sqrt[4]{d}, 1]$} | 78.5 |
| QT-ViT (ours) | 2nd order Taylor expansion | \multicolumn{2}{c}{$\left[\frac{1}{\sqrt{2}}\tilde{K}_r(\phi(\mathbf{x})), \frac{1}{\sqrt{2}}\right]$} | **79.6** |

## 4.2 Ablation Study

In this section, we conduct several ablation studies to further verify the superiority of our proposed quadratic Taylor expansion kernel.

**Results of using different kernels.** We compare the results of using quadratic Taylor expansion kernel with other kernels used in various linear attention vision transformers including EfficientViT [3] which uses ReLU kernel, Hydra attention [2] that utilizes cosine kernel, PolyNL [1] with the mean kernel, AFT-Simple [41] that proposes different kernels for query and key, angular kernel expansion [39] and first order Taylor expansion [7]. All methods use the same training settings and network architecture except the kernel used for computing the linear self-attention. The baseline method uses the original self-attention operation with $\mathcal{O}(N^2 d)$ computational complexity and is used as a strong baseline for comparison.

As the results shown in Tab. 2, the proposed quadratic (2nd order) Taylor expansion outperforms all other competitors which demonstrates the effectiveness of the proposed method. For example, we achieve a 1.1% better top-1 accuracy compared to the linear (1st order) Taylor expansion kernel, 0.2% better than ReLU, 0.5% better than cosine, 0.8% better than mean, 0.7% better than sigmoid & softmax and 0.5% better than the angular kernel.

**Ablation on reducing the time complexity of the Kronecker product.** In Sec. 3.2, we mentioned several ways of reducing the computational burden of the Kronecker product including:

*Method 1:* applying a pooling function on the input vector $\mathbf{y} = pool(\mathbf{x}) \in \mathbb{R}^{d/p}$;

Table 3: Ablation on reducing the time complexity of the Kronecker product. The experiments are conducted using the QT-ViT-1 model on the ImageNet-1k dataset.

| Method | Hyper-param | Time Comp. | Params (M) | FLOPs (G) | Top-1 Acc (%) |
|---|---|---|---|---|---|
| baseline | - | $\mathcal{O}(Nd^3)$ | 9.4 | 0.65 | 79.7 |
| 1 | p=2 | $\mathcal{O}(Nd^3/p^2)$ | 9.4 | 0.55 | 79.3 |
|  | p=4 |  | 9.4 | 0.52 | 79.1 |
| 2 | c=2 | $\mathcal{O}(Nd^3/c)$ | 9.4 | 0.58 | 79.4 |
|  | c=4 |  | 9.4 | 0.55 | 79.3 |
|  | c=8 |  | 9.4 | 0.53 | 79.1 |
| 3 | - | $\mathcal{O}(Nd^2)$ | 9.4 | 0.52 | 61.8 |
| 4 (ours) | - | $\mathcal{O}(Nd^2)$ | 9.4 | 0.52 | **79.6** |

*Method 2:* dividing the input vector into $c$ chunks, compute the Kronecker product within each chunk and concatenate them together to derive the final output;

*Method 3:* randomly preserving $d$ items from $d^2$ quadratic elements in $\hat{K}_r(\phi(\mathbf{x}))$ (Eq. 10);

*Method 4:* using the self-multiplication terms to represent all quadratic terms to derive the compact version of the original Kronecker product $\tilde{K}_r(\phi(\mathbf{x}))$ (Eq. 11).

The classification results of using different methods mentioned above are shown in Tab. 3, in which the baseline method computes the original Kronecker product and is used to compare with other efficient methods. We can see that reducing input dimension with the pooling function (method 1) or dividing it into chunks (method 2) are not efficient enough since $p$ and $c$ are small compared to the dimension $d$ of the vector and has a higher time complexity. Besides, the classification performances are not satisfying because too much information is lost. Randomly selecting quadratic items (method 3) is computationally friendly but has sub-optimal performance. The proposed compact version of the original Kronecker product (method 4) performs best among all the methods which indicates that using the self-multiplication terms to represent quadratic terms is enough to preserve the information in the output of the Kronecker product.

### 4.3 Visualization

We plot the results of the self-attention maps from the last block given a specific query (column 1, marked as red on the original images) using different attention methods including first-order Taylor expansion [7] (column 2), ReLU non-linearity function [3] (column 3) and the quadratic Taylor expansion used in the proposed QT-ViT (column 4). We can see that the proposed QT-ViT can exhibit a more focused and sharper response on attention feature maps. Furthermore, given a query vector, QT-ViT captures reasonable features on the feature map more accurately compared to the competitors. For example, QT-ViT concentrates on both ears of the dog given the query on the left ear of the dog. We can intuitively observe the advantages of QT-ViT from Fig. 2.

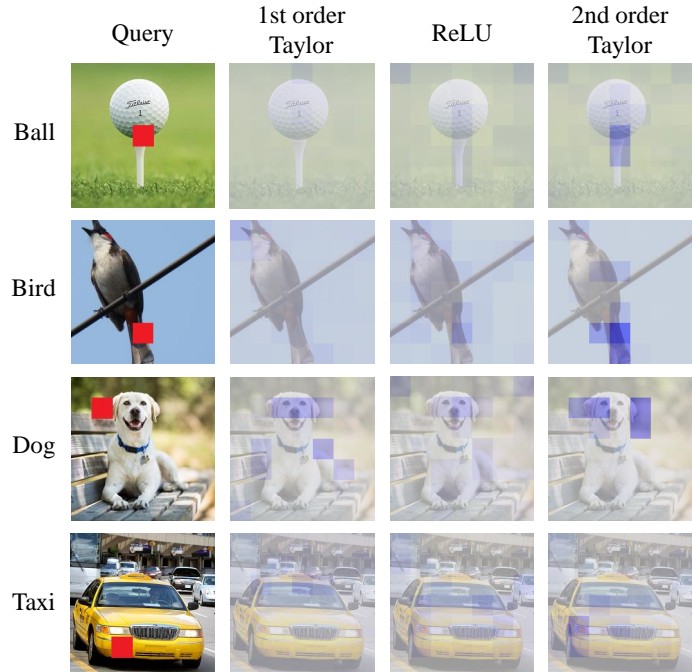

Figure 2: Attention maps from different linear attention methods including the first-order Taylor expansion, ReLU non-linearity function and the second-order Taylor expansion (ours).

## 5   Conclusion

In this paper, we propose a new linear attention method to approximate the usage of softmax self-attention in the original vision transformer models. By conducting quadratic Taylor expansion of the similarity measurement function with the help of the Kronecker product, we can successfully decompose the similarity function into the product of two kernel embeddings while reserving high-order information and maintaining the effectiveness of the original self-attention. Furthermore, we propose a fast approximation algorithm to accelerate the computation of the Kronecker product and reduce the time complexity from $\mathcal{O}(Nd^3)$ to $\mathcal{O}(Nd^2)$ without much loss of information. We conduct experiments on the proposed QT-ViT models using the benchmark dataset ImageNet-1k,

and the results show that we can achieve a better accuracy-efficiency trade-off compared to other state-of-the-art transformers and CNNs.

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

## A    Object Detection on COCO 2017 Dataset

We conduct experiments on the COCO 2017 dataset to further validate the effectiveness of the proposed QT-ViT models. The COCO 2017 dataset has 118K training images, 5K validation images and 20K test-dev images. We use the mask R-CNN [18] as the object detection framework, EfficientViTs [3] and the proposed QT-ViTs as the backbone models. The experimental results are shown in the following table. We can see that the proposed QT-ViT has better mAP than the state-of-the-art model EfficientViT under the same training setting.

Table 4: Experimental results on COCO 2017 dataset using different backbones.

| Backbone | AP | $AP_{50}$ | $AP_{75}$ | Params (M) |
|---|---|---|---|---|
| EfficientViT-B1 | 39.1 | 58.0 | 41.8 | 57.6 |
| QT-ViT-1 | **39.3** | **58.2** | **42.1** | 57.9 |
| EfficientViT-B2 | 40.8 | 59.5 | 44.3 | 68.0 |
| QT-ViT-2 | **41.1** | **59.7** | **44.7** | 68.5 |
| EfficientViT-B3 | 42.3 | 60.6 | 45.5 | 92.1 |
| QT-ViT-3 | **42.6** | **60.9** | **45.9** | 93.1 |

In the following table, we show the results of using absolute positional embedding (APE) on object detection. Note that APE has little impact on the latency, FLOPs, and top-1 accuracy for image classification tasks thus we do not show the corresponding results in the main section.

Table 5: Experimental results on COCO 2017 dataset using different backbones.

| Backbone | AP | $AP_{50}$ | $AP_{75}$ |
|---|---|---|---|
| QT-ViT-1 w/ APE | 39.3 | 58.2 | 42.1 |
| QT-ViT-1 w/o APE | 39.2 | 58.2 | 42.0 |
| QT-ViT-2 w/ APE | 41.1 | 59.7 | 44.7 |
| QT-ViT-2 w/o APE | 41.0 | 59.7 | 44.6 |
| QT-ViT-3 w/ APE | 42.6 | 60.9 | 45.9 |
| QT-ViT-3 w/o APE | 42.5 | 60.8 | 45.8 |

## B    Semantic Segmentation on ADE20K dataset

We further verify the effectiveness of the proposed QT-ViT on the semantic segmentation task using the ADE20K dataset, which contains 20K training images from 150 semantic categories, 2K validation images and 3K test-dev images. UperNet [35] is used as the framework for the experiments. As shown in the table below, using QT-ViTs as the backbone models achieve better mIoU on the ADE20K dataset than using the EfficientViTs as backbones. The results show that the proposed QT-ViT performs well on the semantic segmentation task.

## C    Memory Footprint

The masked output of the original softmax attention has been shown to be useful in previous studies such as Vitality [7] and Castling-ViT [39]. It is also shown to be useful in our method and the experiments using QTViT-1 on the ImageNet dataset are shown below. However, it requires more GPU memory during training, which is not suitable for training large models. Thus, we do not use this strategy in our method. For the integrity of the paper, we still list the results of using original softmax in the following table, the experiments are conducted with QT-ViT-1 on ImageNet dataset.

Table 6: The effectiveness of APE.

| Backbone | mIoU | mAcc | Params (M) |
|---|---|---|---|
| EfficientViT-B1 | 32.8 | 45.3 | 32.5 |
| QT-ViT-1 | **33.2** | **45.6** | 32.8 |
| EfficientViT-B2 | 35.8 | 49.0 | 43.7 |
| QT-ViT-2 | **36.3** | **49.4** | 44.3 |
| EfficientViT-B3 | 38.0 | 51.0 | 68.8 |
| QT-ViT-3 | **38.5** | **51.5** | 69.7 |

Table 7: The impact of using original softmax attention during training.

| Method | GPU memory required per GPU during Training (GB) | Top-1 Acc (%) |
|---|---|---|
| w/o original softmax | 13.9 | 79.6 |
| w/ original softmax | 15.8 (+13.7%) | 79.8 |

## D   Broader Impact

This paper presents work whose goal is to advance the field of Machine Learning. There are many potential societal consequences of our work, none of which we feel must be specifically highlighted here.

