# OpenReview forum: "QT-ViT: Improving Linear Attention in ViT with Quadratic Taylor Expansion"
_NeurIPS.cc/2024/Conference — NeurIPS 2024 poster_

### Official Review · Reviewer_LZeB · 2024-07-04

**Soundness:** 4
**Presentation:** 3
**Contribution:** 4
**Rating:** 7
**Confidence:** 5

**Summary:**

In this paper, the authors propose QT-ViT models, which improve the traditional linear self-attention methods by using a second-order (quadratic) Taylor expansion to approximate the original softmax attention and then accelerate this process using a fast approximation algorithm reducing computational complexity from $O(n^2d)$ to $O(nd^3)$ and further to $O(nd^2)$. This method leverages the properties of quadratic expansion for better performance while maintaining the speed of linear approximation. Extensive experiments on image classification, as well as object detection and semantic segmentation tasks, demonstrate that QT-ViT models achieve state-of-the-art accuracy-speed trade-offs, surpassing previous methods across various model sizes.

**Strengths:**

+ The paper introduces a novel method as it combines the benefits of quadratic expansion with a fast approximation algorithm, offering a fresh perspective on improving attention mechanisms without relying on knowledge distillation or high-order attention residuals.

+ The paper provides a clear theoretical foundation for their method, explaining how the quadratic Taylor expansion and Kronecker product are utilized to reduce computational complexity. The theoretical analysis is solid and sound.

+ The experiments on image classification, detection, and segmentation both show the effectiveness of the proposed method.

+ The paper is well-written and structured.

**Weaknesses:**

+ In Table 1, the top 1 acc of the proposed QT-ViT models over EfficientViT appears to diminish. This trend raises concerns about the scalability and robustness of QT-ViT models as they are scaled up. The paper should include a detailed analysis of this trend.

+ While Figure 1 effectively visualizes the latency of the models, the paper does not include this crucial metric in Table 1, The paper should include latency metrics in Table 1 to provide a comprehensive comparison of the models.

+ Table 2 presents the results of using different kernels, but it is unclear whether all these kernels have the same time complexity. The paper should clearly state whether the time complexities of each kernel in Table 2 are the same. If not, the time complexity of each method should be added in Table 2.

+ In Eq.11. the authors state that using the self-multiplication terms can effectively represent all quadratic terms. Could the authors provide more details for this finding?

+ It seems that the improvement in segmentation is larger than in image classification and object detection. It is better to provide some analysis.

**Questions:**

Please see the weakness. The instability of accuracy is my major concern of this method.

---

> ### Author Rebuttal · Authors · 2024-08-05
>
> **Q1: In Table 1, the top 1 acc of the proposed QT-ViT models over EfficientViT appears to diminish. This trend raises concerns about the scalability and robustness of QT-ViT models as they are scaled up. The paper should include a detailed analysis of this trend.**
>
> A1: First of all, the difficulty of increasing the classification accuracy from a baseline of 79% is different from that of 85%. The better the baseline method is, the more difficult to further improve its performance.
>
> Secondly, similar to EfficientViT B series and L series, QTViT 1$\sim$3 and 4$\sim$6 use different architectures as their baseline models. Thus, the proportion of attention operation is different in these two settings. Specifically, the ratios of FLOPs of attention blocks in QTViTs are shown below and we can see that the ratios in QTViT 4$\sim$6 are much smaller than those in 1$\sim$3. Thus, the proposed method will have a smaller impact on QTViT 4$\sim$6, since we only modify the attention operation in the model. This is the main reason that QTViT 1$\sim$3 has a better performance over the baseline method than QTViT 4$\sim$6 over the baseline method.
>
> |model |ratio of FLOPs of attention blocks|
> |-|-|
> |QTViT-1|59.2%|
> |QTViT-2|62.1%|
> |QTViT-3|65.6%|
> |QTViT-4|25.4%|
> |QTViT-5|25.7%|
> |QTViT-6|25.6%|
>
> **Q2: While Figure 1 effectively visualizes the latency of the models, the paper does not include this crucial metric in Table 1, The paper should include latency metrics in Table 1 to provide a comprehensive comparison of the models.**
>
> A2: Thank you for your advice. We will include the latency metrics in Table 1.
>
> **Q3: Table 2 presents the results of using different kernels, but it is unclear whether all these kernels have the same time complexity. The paper should clearly state whether the time complexities of each kernel in Table 2 are the same. If not, the time complexity of each method should be added in Table 2.**
>
> A3: Sorry for the unclarity. We have mentioned that we compare our method with other kernels used in various linear attention vision transformers in Line 212. These kernels are the kernel functions introduced in Section 2.3. Since they are all used for linear attention, the time complexity of each kernel in Table 2 is all the same and is O(Nd^2). We will further illustrate this in the final version of the paper.
>
> **Q4: In Eq.11. the authors state that using the self-multiplication terms can effectively represent all quadratic terms. Could the authors provide more details for this finding?**
>
> A4: In fact, as Table 3 shows, we have tried several different methods to reduce the number of quadratic terms, and using the self-multiplication terms yields the best result among all of the methods. Generally speaking, the quadratic terms can be viewed as a square matrix $\bf M$ in which each element ${\bf M}_{ij}$ represents the multiplication result of $x_i$ and $x_j$. Thus, the self-multiplication terms can be viewed as the elements on the diagonal of the matrix which can represent the whole matrix to some extent. Besides, we expand the self-multiplication results with respect to the number of quadratic terms and use a learnable parameter $\alpha$ to further adjust the ratio. Thus, the overall method can represent all quadratic terms.
>
> **Q5: It seems that the improvement in segmentation is larger than in image classification and object detection. It is better to provide some analysis.**
>
> A5: In our research, we introduce a novel linear self-attention using quadratic Taylor expansion. Our experimental results on image classification tasks using the ImageNet 1K dataset show the efficacy of our method.
>
> While image classification focuses on assigning a single label to an entire image, segmentation is the classification task on the pixel level. Due to the similarity between these tasks, our method also has a good performance on segmentation.
>
> For the object detection task, treating QT-ViT-1 as an example, although achieving a modest performance improvement compared to the semantic segmentation task, there is a smaller ratio of parameter (57.6M->57.9M, +0.5%) increase compared to the semantic segmentation task with parameters from 32.5M to 32.8M (+0.9%), which is reasonable.
>
> The explanation above will be added in the final version of the paper.

---

> > ### Comment · Reviewer_LZeB · 2024-08-08
> >
> > I appreciate your response. I noticed that some model series, such as Poolformer, are listed in Table 1 but are not shown in Figure 1. Could you please explain this?

---

> > > ### Author Response · Authors · 2024-08-09
> > > **Respond to Reviewer LZeB**
> > >
> > > Thanks for your question. In fact, we have tested the latencies of Poolformer series and the results are shown below.
> > >
> > > |model|latency (ms)|Top-1 Acc (%)|
> > > |-|-|-|
> > > |s12|3.78|77.2|
> > > |s24|7.12|80.3|
> > > |s36|9.91|81.4|
> > > |m36|12.67|82.1|
> > > |m48|16.56|82.5|
> > >
> > > The results of Poolformer series are not as strong as other series and including them in Figure 1 will cause other models to be crowded together, making it difficult for readers to distinguish the strengths and weaknesses. Thus, we decide not to include Poolformer series in Figure 1. Other model series listed in Table 1 but not shown in Figure 1 are due to the similar reason.
> > >
> > > We will add a 'Latency' column in Table 1 in the final version of our paper.

---

> > > > ### Comment · Reviewer_LZeB · 2024-08-13
> > > >
> > > > I am satisfied with the authors' rebuttal. Since they have addressed all of my concerns, I decide to keep my score positive.

---

### Official Review · Reviewer_dQG7 · 2024-07-08

**Soundness:** 3
**Presentation:** 4
**Contribution:** 3
**Rating:** 7
**Confidence:** 5

**Summary:**

This paper proposes a novel method to compute the kernel function in linear attention. They decompose the softmax attention with Tayler expansion and utilize the first two items to approximate the exponential function. The Kronecker product is used to decompose the quadratic Taylor expansion into two kernel functions, and the self-multiplication terms in the output of the kernel are used to replace the quadratic terms for fast inference. The experimental results on multiple CV tasks show the effectiveness and efficiency of the proposed method.

**Strengths:**

- The use of Kronecker product to realize the second-order Taylor expansion is smart and enlightening.

- The fast approximation algorithm is effective in reducing the time complexity.

- The QTViTs achieve a new Parato-front based on the accuracy and speed trade-off.

- The paper is well-written and easy to understand.

**Weaknesses:**

- What is the dimension of $\alpha$, $\beta$ and $\gamma$ used in Eq.11? Are there any ablation studies using different parameters?

- I notice that you replace $K_r(\phi(x))$ in Eq.8 with Eq.11, and you use $\gamma$ to represent the constant term. Then, could the constant term $1/\sqrt{2}$ in Eq.11 be merged with $\gamma$? What about the scaling factor $1/\sqrt{2}$ in Eq.8?

- There are 6 models with different model sizes in Tab.1. Why do you plot only 3 models in Fig.1?

**Questions:**

- You have mentioned that you do not necessitate the masked output of the original softmax attention during training as the previous methods do. I wonder if this can be added back to get further performance gain on QTViT?

- You mentioned that QTViT can exhibit a more focused and sharper response around Fig.2. Is there any insight about this phenomenon?

- In Tab.1, the performance gains of QT-ViT 4$\sim$6 to the SOTA are more marginal compared to QT-ViT 1$\sim$3 to the SOTA. Are there any explanations for this phenomenon?

Overall, I would like to see the accuracy and speed trade of the other 3 models to make sure that the proposed method can surpass EfficientViT on all different model sizes.

---

> ### Author Rebuttal · Authors · 2024-08-05
>
> **Q1: What is the dimension of $\alpha$, $\beta$ and $\gamma$ used in Eq.11? Are there any ablation studies using different parameters?**
>
> A1: $\alpha$, $\beta$, and $\gamma$ are all scalars as shown in Line 164 in the original paper. All of them are learnable parameters and are initially set to 1s in the experiments. We further conduct some ablation studies by setting one of them to 0 initially and learning the three parameters normally. The results are shown below (conducted on QTViT-1 on ImageNet dataset).
>
> |$\alpha$|$\beta$|$\gamma$|Top1 Acc (%)|
> |-|-|-|-|
> |1|1|1|79.6|
> |0|1|1|79.3|
> |1|0|1|79.4|
> |1|1|0|79.0|
>
> We can see that the lack of each part in the initialization will influence the performance, and the constant term is the most important term among them. This is reasonable since the constant term is the most important basis for the Fourier series and a proper value can guarantee the approximation to always be greater than 0 which is one of the properties of softmax function.
>
> **Q2: I notice that you replace $K_r(\phi(x))$ in Eq.8 with Eq.11, and you use $\gamma$ to represent the constant term. Then, could the constant term $1/\sqrt{2}$ in Eq.11 be merged with $\gamma$? What about the scaling factor $1/\sqrt{2}$ in Eq.8?**
>
> A2: This is a really good question. Yes, we merge the scaling factor and constant term in Eq.8 with $\alpha$ and $\gamma$ in Eq.11. Thus, the real initialization of $\alpha$, $\beta$, and $\gamma$ after merging are $1/\sqrt 2$, $1/\sqrt 2$ and $\sqrt 2$.
>
> **Q3: There are 6 models with different model sizes in Tab.1. Why do you plot only 3 models in Fig.1?**
>
> A3: Note that the baseline of our method is the EfficientViT, and they plot only EfficientViT L1$\sim$L3 in their paper. We are following the same rule.
>
> Specifically, QTViT 1$\sim$3 corresponds to EfficientViT B1$\sim$B3 and QTViT 4$\sim$6 corresponds to EfficientViT L1$\sim$L3. For the B series and L series in EfficientViT, they use different backbones and thus achieve different Parato-fronts. The L series has a better Parato-front than the B series. We have a similar result that our QTViT 4$\sim$6 has a better Parato-front than QTViT 1$\sim$3. Thus, we only plot QTViT 4$\sim$6 in our original paper.
>
> To better illustrate this, we show all 6 different models in **Figure 2 in the rebuttal PDF**. Note that although QTViT 4$\sim$6 is better than 1$\sim$3, all six models have better accuracy-speed trade-offs than the EfficientViT series.
>
> If you feel it is necessary, we can add QTViT 1$\sim$3 back to Fig.1 in the final version of the paper.
>
> **Q4: You have mentioned that you do not necessitate the masked output of the original softmax attention during training as the previous methods do. I wonder if this can be added back to get further performance gain on QTViT?**
>
> A4: This is a good question. The masked output of the original softmax attention has been shown to be useful in previous studies such as [1] and [2]. It is also shown to be useful in our method and the experiments using QTViT-1 on the ImageNet dataset are shown below. However, it requires more GPU memory during training, which is not suitable for training large models. Thus, we do not use this strategy in our paper.
>
> |Method|GPU memory required per GPU during training|Top1 Acc (%)|
> |-|-|-|
> |w/o original softmax|13.9 GB|79.6|
> |w/ original softmax| 15.8 GB (+13.7%)| 79.8|
>
>
> [1] Vitality: Unifying low-rank and sparse approximation for vision transformer acceleration with a linear Taylor attention. HPCA, 2023.
>
> [2] Castling-vit: Compressing self-attention via switching towards linear-angular attention at vision transformer inference. CVPR, 2023.
>
> **Q5: You mentioned that QTViT can exhibit a more focused and sharper response around Fig.2. Is there any insight about this phenomenon?**
>
> A5: Compared to linear Taylor attention and ReLU attention, our proposed method has a quadratic term, and thus can have a sharper response on input features with larger values, which means that QTViT concentrates more on important input features.
>
> **Q6: In Tab.1, the performance gains of QT-ViT 4$\sim$6 to the SOTA are more marginal compared to QT-ViT 1$\sim$3 to the SOTA. Are there any explanations for this phenomenon?**
>
> A6: First of all, the difficulty of increasing the classification accuracy from a baseline of 79% is different from that of 85%. The better the baseline method is, the more difficult to further improve its performance.
>
> Secondly, similar to EfficientViT B series and L series, QTViT 1$\sim$3 and 4$\sim$6 use different architectures as their baseline models. Thus, the proportion of attention operation is different in these two settings. Specifically, the ratios of FLOPs of attention blocks in QTViTs are shown below and we can see that the ratios in QTViT 4$\sim$6 are much smaller than those in 1$\sim$3. Thus, the proposed method will have a smaller impact on QTViT 4$\sim$6, since we only modify the attention operation in the model.
>
> |model |ratio of FLOPs of attention blocks|
> |-|-|
> |QTViT-1|59.2%|
> |QTViT-2|62.1%|
> |QTViT-3|65.6%|
> |QTViT-4|25.4%|
> |QTViT-5|25.7%|
> |QTViT-6|25.6%|

---

> > ### Comment · Reviewer_dQG7 · 2024-08-09
> > **Thanks**
> >
> > Thanks for the author’s reply. I have some additional questions. Do you use KD method in the table in A4? How many GPUs do you use during training?

---

> > > ### Author Response · Authors · 2024-08-09
> > > **Respond to Reviewer dQG7**
> > >
> > > Thanks for your question.
> > >
> > > We do not use KD method in the table in A4. In fact, adding KD method will largely increase the GPU memory (at least double the GPU memory usage). We use 8 GPUs during training. Each AMD MI250 GPU has a maximum GPU memory of 64GB.
> > >
> > > It is still enough for training QTViT-1 with the help of original softmax attention and KD method, but it will cause OOM issue when training larger models such as QTViT-3. Thus, it shows the advantage of our method that can derive a good performance without utilizing masked softmax attention output or the KD method.

---

> > > > ### Comment · Reviewer_dQG7 · 2024-08-13
> > > >
> > > > Thanks for the author's reply. I am glad to see that the author provides the accuracy-speed trade-off on all 6 different models, and the results show a consistent improvement. Other concerns are also addressed. Thus, I would like to increase my score from 6 to 7.

---

### Official Review · Reviewer_QZCU · 2024-07-10

**Soundness:** 2
**Presentation:** 3
**Contribution:** 2
**Rating:** 5
**Confidence:** 4

**Summary:**

This paper introduces QT-ViT models, which enhance linear self-attention using quadratic Taylor expansion. The similarity function is decomposed into the product of two kernel embeddings via the Kronecker product. By employing a fast approximation algorithm, the computational cost is reduced while maintaining overall performance. Experiments on ImageNet Classification with various model sizes show consistent improvement over the baseline EfficientViT method. Additionally, visualizations demonstrate that the QT-ViT model has a more focused attention feature map.

**Strengths:**

- S1. The paper is well-organized and easy to follow.
- S2. The overall idea of improving linear attention with quadratic Taylor expansion and Kronecker product is sound.
- S3. An extensive ablation study is conducted to show the performance of different variants of the Kronecker product.

**Weaknesses:**

- W1. The overall improvement is incremental. Compared to the EfficientViT baseline, the performance is almost the same under the same computation budget across different model sizes and tasks. For ImageNet classification, it would be helpful to include results on ImageNet-v2 and ImageNet-real to demonstrate the robustness and consistency of the improvement.

- W2. The image resolution is not listed in the experiment section. Performance on different image resolutions should also be reported and compared with other methods. When using a much larger image resolution (and more patch tokens), how are the computation cost and latency affected?

- W3. It would be better to include the detection and segmentation results in the main text to support the effectiveness of the proposed QT-ViT across different tasks.

- W4. The network architecture should be detailed, and additional components like absolute positional embedding should be ablated.

- W5. The mIoU performance of EfficientViT-B3 is reported as 49.0 on the ADE20k dataset in the original paper, which is inconsistent with the 38.0 reported in this paper.

**Questions:**

- The author should provide more implementation details about the network, experimental setting, and comparison with other methods.

**Limitations:**

- The latency is tested on an AMD GPU, and I'm not sure if there is any difference on the implementation side. Will the performance gain be consistent on other type of devices?

---

> ### Author Rebuttal · Authors · 2024-08-05
>
> **Q1: Overall improvement is incremental. Add results on ImageNetv2 and real.**
>
> A1: EfficientViT is currently the state-of-the-art method according to the accuracy-efficiency trade-off. Also, note that the difficulty of increasing the classification accuracy from a baseline of 79% is different from that of 85%. The better the baseline method is, the more difficult to further improve its performance. We achieve a new SOTA accuracy-efficiency trade-off under different model sizes.
>
> Note that our QTViTs 1$\sim$3 and 4$\sim$6 follow the architectures of EfficientViT B and L series which use different architectures as their baseline models, and QTViTs 1$\sim$3 have more obvious improvements than QTViTs 4$\sim$6. One of the reasons is mentioned above. Another reason is that the ratios of FLOPs of attention blocks in QTViTs 4$\sim$6 are much smaller than those in 1$\sim$3. Thus, the proposed method will have a smaller impact on QTViTs 4$\sim$6, since we only modify the attention operation in the model. The proportions of the attention operation are shown in the following table.
>
> |model |ratio of FLOPs of attention blocks|
> |-|-|
> |QTViT-1|59.2%|
> |QTViT-2|62.1%|
> |QTViT-3|65.6%|
> |QTViT-4|25.4%|
> |QTViT-5|25.7%|
> |QTViT-6|25.6%|
>
> For the robustness and consistency of the improvement, we directly use the pre-trained QTViTs and EfficientViTs checkpoints and conduct inference on ImageNet-V2 and ImageNet-ReaL. The results are shown below, and we can see that the proposed QTViTs can consistently outperform EfficientViTs, which shows the robustness and consistency of our method.
> |models|ImageNet|ImageNet-V2 |ImageNet-ReaL|
> |-|-|-|-|
> |QTViT-1|**79.57**|**75.37**|**85.61**|
> |EfficientViT-B1|79.38|75.04|85.32|
> |QTViT-2|**82.46**|**78.21**|**87.32**|
> |EfficientViT-B2|82.09|77.92|86.98|
> |QTViT-3|**83.93**|**80.04**|**88.31**|
> |EfficientViT-B3|83.47|79.01|88.11|
>
> **Q2: Larger image resolution, and the computation cost and latency.**
>
> A2: We use image resolution of 224 by default. The experimental results of using other image resolutions such as 256 and 228 are shown below, we can see that our QTViT can achieve a consistent improvement over the state-of-the-art method EfficientViT, and the FLOPs and latencies are roughly the same. Experiments on other models and the corresponding discussions will be added in the final version of the paper.
> |resolution|model|FLOPs (G)|latency (ms)|Top-1 Acc (%)|
> |-|-|-|-|-|
> |224|QTViT-1|0.52|1.74|**79.6**|
> |224|EfficientViT-B1|0.52|1.76|79.4|
> |256|QTViT-1|0.68|2.16|**80.1**|
> |256|EfficientViT-B1|0.68|2.13|79.9|
> |288|QTViT-1|0.86|2.28|**80.6**|
> |288|EfficientViT-B1|0.86|2.26|80.4|
>
> **Q3: Move det and seg results in the main text.**
>
> A3: Thanks for your suggestion, the experiments are currently in the supplemental section due to the limited pages of the main text. We will move the detection and segmentation results to the main text in the final version.
>
> **Q4: The network architecture should be detailed, and absolute positional embedding should be ablated.**
>
> A4: As shown in lines 176~180, we use exactly the same architecture as EfficientViT, except for changing the kernel function to our quadratic Taylor expansion kernel and adding the absolute positional embedding. In fact, the absolute positional embedding has little impact on the latency, FLOPs, and top-1 accuracy for image classification tasks (with almost the same latency and FLOPs, and <0.05 top-1 accuracy gap on different models). We actually see the difference in object detection, and the results of using absolute positional embedding or not are shown in the following table (APE stands for absolute positional embedding).
>
> |Backbone|AP|AP$_{50}$|AP$_{75}$|
> |-|-|-|-|
> |QTViT-1 w/ APE|39.3|58.2|42.1|
> |QTViT-1 w/o APE|39.2|58.2|42.0|
> |QTViT-2 w/ APE|41.1|59.7|44.7|
> |QTViT-2 w/o APE|41.0|59.7|44.6|
> |QTViT-3 w/ APE|42.6|60.9|45.9|
> |QTViT-3 w/o APE|42.5|60.8|45.8|
>
> The above results will be added to the final version of the paper.
>
> **Q5: Inconsistent mIoU results.**
>
> A5: For the results of semantic segmentation, since EfficientViT did not provide their training details either in their GitHub code or in their paper by the time we submitted our NeurIPS paper, we conducted segmentation experiments based on mmsegmentation and used the same training strategy as Upernet-resnet50. Note that we use exactly the same training strategy to derive the results of the proposed QTViT and EfficientViT, we can surpass them by a margin which shows the priority of the proposed method. It is easy to improve the absolute mIoU value of both EfficientViT and QTViT by adjusting the training strategy, such as increasing the training steps.
>
> As shown in the following table, by merely increasing the training steps, the mIoU results can be largely increased and approach the result of official EfficientViT-B1. In all of the settings, our QTViT can outperform EfficientViT.
>
> |model |training steps|mIoU|
> |-|-|-|
> |QTViT-1|160k|*33.2*|
> |EfficientViT-B1|-|32.8|
> |QTViT-1|320k|**37.6**|
> |EfficientViT-B1|-|37.2|
> |QTViT-1|640k|**41.2**|
> |EfficientViT-B1|-|40.9|
>
>
> **Q6: Provide more implementation details, experimental setting, and comparison with other methods.**
>
> A6: Thanks for your suggestions. The implementation details are already provided in lines 176-180, the experimental settings are shown in the experimental section and we will add the default image resolution in the main text. The experimental results mentioned above will be added to the final version of our paper.
>
> **Q7: The latency on other type of devices.**
>
> A7: We follow the standard procedure to test the latency on AMD GPU, by first converting the .pth pytorch checkpoint file into .onnx file, then conducting a latency test on our device.
>
> We also test the latencies on NVIDIA V100 GPU, and the latency results are shown in **Figure 1 in the rebuttal PDF**. The conclusion is consistent with that of AMD GPU.

---

> > ### Comment · Reviewer_QZCU · 2024-08-12
> > **Thanks for the detailed feedback.**
> >
> > Thank you for the detailed rebuttal and the additional results provided. I appreciate the effort in addressing the issues raised. The consistent improvement over the ImageNet dataset and its variants, along with the consistent latency improvements across different device types, demonstrates the robustness of your approach. Additionally, the extra experiments and ablation studies on detection and segmentation effectively highlight the method's potential.
> >
> > However, I do have one minor concern. If I understand correctly, the resolution tested in A2 is 288 rather than 228. While the results are promising, 288 is still not considered a very large resolution where differences in latency might be more pronounced. Furthermore, the performance gain, though present, remains marginal compared to EfficientViT across various model sizes and computation costs.
> >
> > The overall results suggest that the paper meets the acceptance threshold. Based on the improvements and additional evidence provided, I am willing to increase my score to 5.

---

> > > ### Author Response · Authors · 2024-08-13
> > > **Thank you**
> > >
> > > Thank you very much for your positive decision.
> > >
> > > Yes, the resolution tested in A2 is 288 rather than 228. Since the largest resolution used in EfficientViT-B1 is 288, we use the same setting as EfficientViT-B1 in A2. We further extend the resolution to 320 and 384. The results are shown below, but sorry for the limited discussion time left, we can only provide the FLOPs and latencies results.
> > > |resolution|model|FLOPs (G)|Latency (ms)|
> > > |-|-|-|-|
> > > |320|QTViT-1|1.06|2.44|
> > > |320|EfficientViT-B1|1.06| 2.42|
> > > |384|QTViT-1|1.53|2.63|
> > > |384|EfficientViT-B1|1.53| 2.59|
> > >
> > > Best,
> > >
> > > Authors of paper 275

---

> ### Author Response · Authors · 2024-08-12
> **Are there any further questions?**
>
> Dear reviewer QZCU,
>
> We appreciate your comments and suggestions in the review. All of your concerns in your original review have been addressed, including the overall improvement and results on ImageNetv2 and real, experiments on different image resolutions, ablations on positional embedding, experiments on semantic segmentation, and inference speed comparison on different devices.
>
> We hope you can read the rebuttal and let us know if you have further questions. Thanks for your time.
>
> Best,
>
> Authors of paper 275

---

### Official Review · Reviewer_5zgq · 2024-07-22

**Soundness:** 1
**Presentation:** 2
**Contribution:** 2
**Rating:** 3
**Confidence:** 4

**Summary:**

This paper proposed a new linear complexity sequence modeling strategy for image modeling. To achieve this, the authors first replace the softmax attention with second-order Taylor expansion, then accelerate its computation with a fast approximation algorithm. The effectiveness of the proposed method is validated on the image classification task.

**Strengths:**

Its performance on ImageNet-1K is good.

**Weaknesses:**

1. It is unclear why directly using the first-order Taylor expansion is worse than using the second-order Taylor expansion with linear approximation. The author should provide a section to discuss this.

2. Experiment details are missing. For example, different image resolutions can largely impact image classification performance. The author fails to provide the image resolution in the experiment section.

3. Some mathematical derivation can be moved to the supplemental section. For example, there are some redundant equations in Eq. 4, 6, 7, 9.

4. There is no actual training/inference speed comparison. Counting flops can be misleading in some cases. I would like to know the actual speed instead of flops.

5. Missing linear attention vision backbone. For example, VVT (TPAMI 23).

6. For image modeling, object detection, and semantic segmentation are also important. I do not know why the author decided to move these two tasks into the supplemental section. However, the semantic segmentation results are extremely low when compared with other linear image backbones. The author do not provide any comments on this.

**Questions:**

As above.

---

> ### Author Rebuttal · Authors · 2024-08-05
>
> **Q1: It is unclear why directly using the first-order Taylor expansion is worse than using the second-order Taylor expansion with linear approximation. The author should provide a section to discuss this.**
>
> A1: In fact, the disadvantages of first-order Taylor attention are mentioned in the sections introduction (lines 42-44), preliminaries (lines 102-105), and experiments (lines 198-200), and our main idea is trying to resolve the disadvantages.
> Generally speaking, first-order Taylor expansion discards too much information from the original attention and needs to necessitate the KD or high-order attention residuals during training to make up for the performance gap. However, the GPU memory consumption will severely increase which makes these methods unsuitable for training large transformer models. The second-order Taylor expansion utilizes more information in the original attention and can alleviate this problem. The above explanation is shown in lines 108-112.
>
> **Q2: Experiment details are missing. For example, different image resolutions can largely impact image classification performance. The author fails to provide the image resolution in the experiment section.**
>
> A2: We use image resolution of 224x224 by default in our experiments. The experimental results of using other image resolutions such as 256 and 228 are shown below, we can see that our QTViT can achieve a consistent improvement over the state-of-the-art method EfficientViT. Experiments on other models and the corresponding discussions will be added in the final version of the paper.
> |resolution|model|Top-1 Acc (%)|
> |-|-|-|
> |224|QTViT-1|**79.6**|
> |224|EfficientViT-B1|79.4|
> |256|QTViT-1|**80.1**|
> |256|EfficientViT-B1|79.9|
> |288|QTViT-1|**80.6**|
> |288|EfficientViT-B1|80.4|
>
> **Q3: Some mathematical derivation can be moved to the supplemental section. For example, there are some redundant equations in Eq. 4, 6, 7, 9.**
>
> A3: In fact, Eq. 4, 6, 7, 9 are not redundant equations. Eq.4 shows an initial way of decomposing the $exp(\cdot)$ function with second-order Taylor expansion. Eq.6 gives theoretical proof that the square of a dot product can be converted into two Kronecker products followed by a dot product, which is also the core analysis of our method. Eq.7 combines the results of Eq.4 and 6 to prove that we can decompose the similarity function into separate kernel embeddings. Finally, Eq.9 changes the order of the elements in the Kronecker product, which does not influence the result of Eq.7 and can help us generate the approximation algorithm that can reduce the computational cost.
>
> These equations illustrate the main idea of our method and may not be moved to the supplemental section. Otherwise, it will be hard for the readers to understand our method.
>
> **Q4: There is no actual training/inference speed comparison. Counting flops can be misleading in some cases. I would like to know the actual speed instead of flops.**
>
> A4: Figure 1 already shows the inference speed comparison among different methods. We provide inference speed of the models on the AMD Instinct MI250 GPU, which is also shown in Line 192 in the original paper.
>
> **Q5: Missing linear attention vision backbone. For example, VVT (TPAMI 23).**
>
> A5: Thanks for your suggestion. We will add VVT in our final version. We show the comparison between our method and VVT in the following table. We can see that we achieve similar results compared to VVT-small and medium but with much fewer FLOPs. For VVT-tiny and large, we can outperform them by a large margin with much fewer FLOPs and parameters.
>
> |models|parameters|FLOPs|Top-1 Acc (%)|
> |-|-|-|-|
> |QTViT-1|**9.4**|**0.52**|**79.6**|
> |VVT-tiny|12.9|3.0|79.2|
> |-|-|-|-|
> |QTViT-2|24.9|**1.60**|82.5|
> |VVT-small|25.5|5.6|82.6|
> |-|-|-|-|
> |QTV-T-3|49.7|**3.97**|83.9|
> |VVT-medium|47.9|9.4|83.8|
> |-|-|-|-|
> |QTViT-4|**53.0**|**5.26**|**84.7**|
> |VVT-large|61.8|10.8|84.1|
>
> **Q6: For image modeling, object detection, and semantic segmentation are also important. I do not know why the author decided to move these two tasks into the supplemental section. However, the semantic segmentation results are extremely low when compared with other linear image backbones. The author do not provide any comments on this.**
>
> A6: This is a good question. Due to space limitations, we move the results of object detection and semantic segmentation into the supplemental section. We will put them into the experimental section in the final version of the paper.
>
> For the results of semantic segmentation, since EfficientViT did not provide their training details either in their GitHub code or in their paper by the time we submitted our NeurIPS paper, we conducted segmentation experiments based on mmsegmentation and used the same training strategy as Upernet-resnet50. Note that we use exactly the same training strategy to derive the results of the proposed QTViT and EfficientViT, we can surpass them by a margin which shows the priority of the proposed method. It is easy to improve the absolute mIoU value of both EfficientViT and QTViT by adjusting the training strategy, such as increasing the training steps.
>
> As shown in the following table, by merely increasing the training steps, the mIoU results can be largely increased and approach the result of official EfficientViT-B1. In all of the settings, our QTViT can outperform EfficientViT.
>
> |model |training steps|mIoU|
> |-|-|-|
> |QTViT-1|160k|**33.2**|
> |EfficientViT-B1|-|32.8|
> |QTViT-1|320k|**37.6**|
> |EfficientViT-B1|-|37.2|
> |QTViT-1|640k|**41.2**|
> |EfficientViT-B1|-|40.9|

---

> > ### Comment · Reviewer_5zgq · 2024-08-13
> >
> > 1. The author addressed why using second-order Taylor expansion is better than the first-order Taylor expansion, but did not directly explain why using the first-order Taylor expansion is worse than using the second-order Taylor expansion with linear approximation. This does not resolve my concern in this case.
> >
> > 2. The actual training speed is still missing. Also, the inference speed is measured on an AMD Instinct MI250 GPU. I am not sure if all other methods have been fully optimized on AMD GPUs since other methods were originally implemented for NV GPUs.

---

> ### Author Response · Authors · 2024-08-12
> **Are there any further questions?**
>
> Dear reviewer 5zgq,
>
> We appreciate your comments and suggestions in the review. All of your concerns in your original review have been addressed, including the clarity of second-order Taylor expansion, experiments on different image resolutions, mathematical derivation, inference speed, comparison with VVT and experiments on semantic segmentation.
>
> We hope you can read the rebuttal and let us know if you have further questions. Thanks for your time.
>
> Best,
>
> Authors of paper 275

---

> > ### Comment · Area_Chair_GLPt · 2024-08-12
> > **Let's engage in the reviewer-author disscussion**
> >
> > Dear Rev. 5zgq,
> >
> > We look forward to seeing your comments on the authors' rebuttal, as well as any further clarifications as you may need.
> >
> > Thanks

---

> ### Author Response · Authors · 2024-08-13
> **Rebuttal to reviewer 5zgq**
>
> Thanks for your comments.
>
> 1. If I understand it correctly, you are asking why using our fast approximation algorithm can well represent the second-order Taylor expansion. In fact, as Table 3 shows, we have tried several different methods to reduce the number of quadratic terms, and using the self-multiplication terms yields the best result among all of the methods. Generally speaking, the quadratic terms can be viewed as a square matrix ${\bf M}$ in which each element ${\bf M}_{ij}$ represents the multiplication result of ${\bf x}_i$ and ${\bf x}_j$. Thus, the self-multiplication terms can be viewed as the elements on the diagonal of the matrix which can represent the whole matrix to some extent. Besides, we expand the self-multiplication results with respect to the number of quadratic terms and use a learnable parameter
>  to further adjust the ratio. Thus, the overall method can represent all quadratic terms.
>
> 2. The training speeds of EfficientViT and QTViT are shown below, in which we use a batch size of 1024. The inference speeds of different models on NVIDIA V100 GPU are shown in **Figure 1 in the rebuttal pdf** which is also used to answer Q7 from reviewer QZCU.
>
> |model|training speed (ms/batch)|
> |-|-|
> |EfficientViT-B1|534.5|
> |QTViT-1|538.0|
> |EfficientViT-B2|586.6|
> |QTViT-2|591.5|
> |EfficientViT-B3|756.8|
> |QTViT-3|763.4|
>
> Hope these answers can address your concerns.

---

> > ### Comment · Reviewer_5zgq · 2024-08-13
> >
> > Thanks for your prompt reply.
> >
> > I am digging into the approximation and found a paper [1] that is very similar to the proposed method, i.e., doing a second-order Taylor expansion of exp(x) and then writing it as inner products. I noticed that [1] was first published on arxiv on 28 February 2024 and accepted in ICLR 24. Here is an implementation as well: https://github.com/lucidrains/taylor-series-linear-attention.
> >
> > Would you please provide a detailed discussion of the differences between these two methods? Also, the paper is missing a citation of [1].
> >
> > [1] Arora, S., Eyuboglu, S., Zhang, M., Timalsina, A., Alberti, S., Zinsley, D., ... & Ré, C. (2024). Simple linear attention language models balance the recall-throughput tradeoff. arXiv preprint arXiv:2402.18668.

---

> ### Author Response · Authors · 2024-08-13
> **Rebuttal to reviewer 5zgq**
>
> First of all, this paper directly used the conclusion of 2nd Taylor expansion from paper [a1] at the start of page 7. However, the conclusion in paper [a1] has some problems. For example, in page 6 in paper [a1], they have $d'=1+d+d^2$. However, the dimension of $\varphi(x)$ should be $(d+1)^2+1=2+2d+d^2$ which can be rigorously derived from Eq.8 in our paper. Besides, paper [a1] does not have any theoretical analysis and derivations to explain the dimension $d'$. Thus, we believe that our conclusion is correct.
>
> Secondly, the acceleration of this paper is to utilize the hardware-efficient algorithm based on CUDA, which limits the utilization of its method and can only be accelerated on NVIDIA GPUs and is not able to be accelerated with other devices. The time complexity of their method is still $O(Nd^3)$, and mapping the input to lower dimensions as they used will yield bad results (see method 1 and 2 in Table 3 in our paper). Our method provides a fast approximation algorithm to reduce the time complexity to $O(Nd^2)$ and can be used without limitation.
>
> Thirdly, this paper was published on Arxiv only two months before the deadline of NeurIPS 2024, thus we do not cite this paper or discuss it. We will add these discussions in our final version.
>
> Thank you very much.
>
> [a1] The Hedgehog & the Porcupine: Expressive Linear Attentions with Softmax Mimicry. ArXiv, 2024.

---

> > ### Comment · Reviewer_5zgq · 2024-08-14
> >
> > The issue in [a1] has been discussed in the paper "Linear Transformers with Learnable Kernel Functions are Better In-Context Models". However, the distinction between this paper and your approach is that one uses "concat" while the other uses "add".
> >
> > The contribution of the hardware implementation of different platforms is not enough for a NeurIPS paper.
> >
> > Can you rephrase the main insight or contribution of your paper based on this information?

---

> ### Author Response · Authors · 2024-08-14
> **Are there any further questions?**
>
> Dear reviewer 5zgq,
>
> We appreciate you taking the time to review this paper. Since it is less than 10 hours before the deadline for the discussion phase, we hope that we have addressed all your concerns.
>
> If you still have any questions, we are happy to discuss them with you. Thank you very much.
>
> Best,
>
> Authors of paper 275

---

> ### Author Response · Authors · 2024-08-14
> **Rebuttal to reviewer 5zgq**
>
> The paper [a2] you mentioned does not solve any of the problems in [a1] that I mentioned above. In fact, they do not even cite the paper [a1]. This paper simplifies the second-order Taylor expansion to $\phi(x)=x^2$ as shown on page 5, and uses a linear transformation before applying the kernel function. Thus, their implementation is much simpler than the second-order Taylor expansion since they discard all other terms except for the quadratic term. There is no theoretical analysis to show the relationship between this method and the second-order Taylor expansion.
>
> Thus, compared to this paper [a2], our contributions are concluded in the following:
>
> 1. We improve the linear self-attention using the **correct** quadratic Taylor expansion through rigorous theoretical analysis and derivations, while there is no such analysis in [1, a1, a2] and the conclusions they derive are either incorrect [1, a1] (see the dimension $d’$ I mentioned above) or has little connection to second-order Taylor expansion [a2].
>
> 2. We propose a fast approximation algorithm to reduce the time complexity of our method from $O(Nd^3)$ to $O(Nd^2)$, while they use methods such as reducing the dimension of input through the linear layer or using hardware-efficient algorithm based on CUDA, which have their own drawbacks as we analyzed previously.
>
> 3. These three papers [a1, a2, 1] are all published within 2-3 months before the submission deadline of NeurIPS 2024. Thus it is unfair to judge our novelty based on them according to the policy of the NeurIPS conference.
>
> Besides, the contribution of the hardware implementation of different platforms is about the paper [1] you mentioned above rather than ours.
>
> Hope the above analysis can address your concern.
>
> [a2] Linear Transformers with Learnable Kernel Functions are Better In-Context Models. ArXiv, 2024.

---

> > ### Comment · Reviewer_5zgq · 2024-08-14
> >
> > Well, all linear methods share the same complexity of O(nd^2).
> >
> > The paper begins with a second-order Taylor expansion to reduce the complexity to O(nd^3), and then utilizes a feature map of concat(q^2, q) to further reduce the complexity to O(nd^2).
> >
> > However, the transformation between Taylor expansion to concat(q^2, q) is not smooth.
> >
> > Using concat(q^2, q) is somewhat new. However, I would like to see stronger evidence for its efficacy. The results of other vision tasks are not compelling.
> >
> > Thanks for the quick response and for discussing the related work. Considering the others' ratings, I would like to keep my initial rating as the avg 5.5 reflects the quality of this work.

---

> > > ### Author Response · Authors · 2024-08-14
> > > **Regret to hear that**
> > >
> > > We I regret to hear that you wouldn’t change your score. In fact, simply concat(q^2,q) yield bad result, since the coefficients before each term and the constant term are important to derive our sota results.
> > >
> > > We believe that the simplicity of our implementation is an advantage rather than drawback. The analysis and derivation to the final equation form is the most important part of our method, which is not discussed by other papers before.
> > >
> > > After all, we appreciate your time for reviewing this paper.

---

### Author Rebuttal · Authors · 2024-08-05

The PDF contains the answer to Q7 of Reviewer QZCU that adds the latencies of different models on NVIDIA-V100 GPU, and the answer to Q3 of Reviewer dQG7 that adds the results of QTViT 1-3 and EfficientViT-L series into the figure.

---

### Decision · Program_Chairs · 2024-09-25

**Decision:**

Accept (poster)

**Comment:**

This paper presents a method for improving the prior art of linear self-attention using quadratic Taylor expansion with a fast approximation algorithm. It has been reviewed by several knowledgable reviewers. The majority consensus from the reviewers (dQG7, LZeB and QZCU) was that the proposed fast approximation algorithm is novel and effective. The clarifications and new results presented during the reviewer-author discussion addressed their concerns raised in the initial comments. This meta-reviewer concurs, and recommends **acceptance**.

The reviewer 5zgq had concerns mainly regarding the difference between the proposed method and some related work pointed out during the reviewer-author discussion. The authors and the reviewer 5zgq had very informative discussions. Although the authors did not finally convince the reviewer 5zgq, s/he did mentioned that the proposed concatenation of self-multiplication terms in the fast approximation algorithm is somewhat new. So, this meta-review leans towards looking at the positive aspects of the proposed method.

The authors are encouraged to carefully revise their paper by,
- incorporating the clarifications, discussions, new results and new references;
- providing comparisons on memory footprints that are missing in the submission and the discussions;
- providing more detailed analyses on effects between w/ APE and w/o APE in both image classification tasks and downstream tasks.